



# Field inter-comparison of low-cost sensors for monitoring methane emissions from oil and gas production operations

Vincent M. Torres, David W. Sullivan, Elyse He'Bert, Jarett Spinhirne, Mrinali Modi and David T. Allen

[1]Center for Energy and Environmental Resources, University of Texas, Austin, 78758, United States

*Correspondence to*: David T. Allen (allen@che.utexas.edu)

**Abstract.** Four solar-powered methane sensing systems with remote communication capabilities were tested for nine months at an oil and gas production site in west Texas. Sensor performance was evaluated using single blind certified gas challenges and by comparison with a continuously operated quantum cascade tunable infrared laser differential absorption spectrometer

(QC-TILDAS) system. Dispersion modelling was used to estimate concentrations that would need to be detected to identify continuous and intermittent emission rates of 5-10 kg/hr from oil and gas production sites within ~50-100 m of the sensors, and these concentration thresholds were used in establishing performance criteria for the sensors. The four sensors demonstrated sufficient precision to allow for detection of emission rates of 5-10 kg/hr and had data capture rates that exceeded 80% during the 9 months of operation. One sensor had a 100% data capture rate, despite severe weather conditions and

extended local electrical power losses. These results demonstrate that multiple commercially available sensing systems are suitable for long term methane emission monitoring in remote oil and gas production regions.

## 1 Introduction

Methane is a greenhouse gas with a global warming potential 28−34 times higher than carbon dioxide over a 100 year period (IPCC, 2014). In the United States, oil and natural gas systems are a major source of methane emissions (NASEM, 2018), and

of the methane emissions attributed to oil and gas supply chains, 40-60% have been attributed to production sites (U.S. EPA, 2021a; Alvarez, et al., 2018). Short duration measurements, lasting from seconds to minutes, using multiple sensing systems and measurement platforms, have been used to quantify methane emissions from natural gas production, either at the level of individual equipment on production sites (Allen, et al., 2013; 2015a,b), or as site-wide totals (Allen, et al., 2013; Yacovitch, et al., 2014, 2015; Rella, et al., 2015; Lan, et al., 2015; Lyon, et al., 2016; Zavala, et al., 2018; Schwietzke, et al., 2019). Short

duration measurements provide instantaneous snapshots of emissions, however, since many emission sources in upstream oil and gas operations are intermittent, short duration measurements may not detect all emissions from a site or may observe an intermittent emission that is then interpreted as persistent. In addition, since most short-term measurements are deployed on a monthly, quarterly, semi-annual or annual basis, emissions that develop between measurements could persist undetected until the next scheduled measurement. If these emission rates are large, total emissions could be dominated by sources that

develop between scheduled measurements. These limitations of periodic, short duration measurements have driven interest in





continuous monitoring of emissions, using networks of sensors. The primary advantage of a continuous monitoring network is that it may be able to detect methane emissions much more quickly than detection methods based on short sampling times that are periodically repeated. The disadvantage of such networks is the cost of deploying the large numbers of sensors that might be required to detect emissions.


The goals of this work were to establish performance criteria for continuous methane sensor systems, and to test sensor systems in field inter-comparisons. The sensors were tested for their ability to accurately measure ambient concentrations of methane over concentration ranges and with time responses that would detect typical emission rates associated with oil and gas production. Dispersion modeling was used to transform emission detection targets into criteria for sensor precision and

temporal resolution. A nine month field trial in the Permian Basin of west Texas was used to assess sensor performance.

## 2 Methods

### 2.1 Sensor performance criteria

The emission detection target established for this work was that a sensor would be able to detect an emission rate of 5-10 kg/hr at a distance of ~50-100 m from the emission source. This emission threshold was chosen based on previous studies of oil and

gas production regions in the United States. For example, Zavala, et al. (2017) analysed methane emission rate data collected from a large number of oil and gas production sites in the Barnett Shale production region in north central Texas and concluded that sites with instantaneous emission rates above 26 kg/hr (1% of sites) accounted for 44% of methane emissions. Sites with instantaneous emissions greater than 10 kg/hr (4% of sites) accounted for approximately 68% of emissions. Based on data such as these, emissions in excess of 5-10 kg/hr should account for a large fraction of total emissions in many oil and gas

production regions. A target of 5-10 kg/hr is also consistent with proposed requirements for advanced measurement technologies for methane emissions from oil and gas production, published by the U.S. Environmental Protection Agency (EPA, 2021b).

Since many emission sources in oil and gas operations are intermittent (Allen, et al., 2017) with durations of one minute or

less, advanced sensing systems for methane emissions require a temporal resolution of one minute or less. The emission rate (5-10 kg/hr detectable at a distance of 50-100 m) and temporal resolution requirements (one minute resolution) for sensors were converted into requirements for precision using dispersion modelling. Since the continuous monitors were deployed in the Permian Basin oil and gas production region, meteorological data required for the dispersion modelling were assembled at one minute resolution, from the Permian Basin. Four time periods of one-week duration, from four calendar quarters, were

identified to be representative of meteorological conditions during 2019. The year was first divided into four quarters. Wind speeds and wind directions observed in each 7-day period within the quarter were then evaluated against the distribution of wind speeds and directions seen in the entire quarter. The weeks that captured a reasonable representation of the range and





frequency of wind speeds and wind directions observed during each quarters were selected as representative weeks. The dataset employed for this selection was obtained from a ground-based monitoring station in the Permian Basin (Continuous Ambient

Monitoring Station, CAMS 47), operated by the Texas Commission on Environmental Quality (TCEQ, 2020). As shown in Figure 1 and 2, the week-long periods were found to be representative of the annual and quarterly variability in meteorology.

**Figure 1. Wind rose diagrams, showing wind speed frequency data, by wind direction for the four representative weeks and for observational annual average data.**

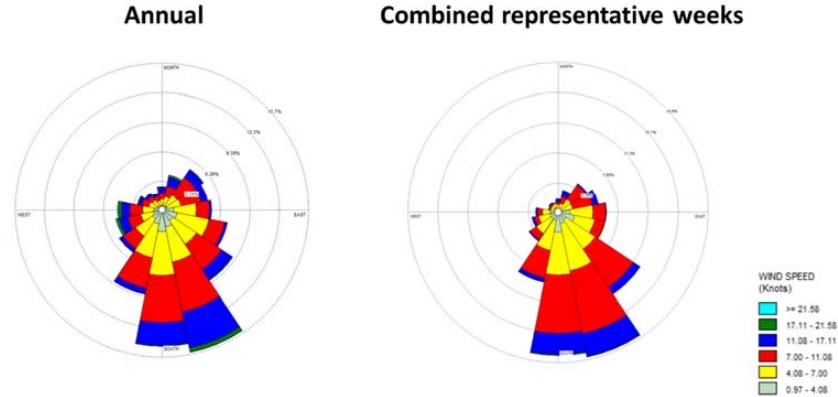






**Figure 2. Wind roses for each quarter (left) and representative week (right) for quarters 1 (a), 2 (b), 3 (c), and 4(d).**

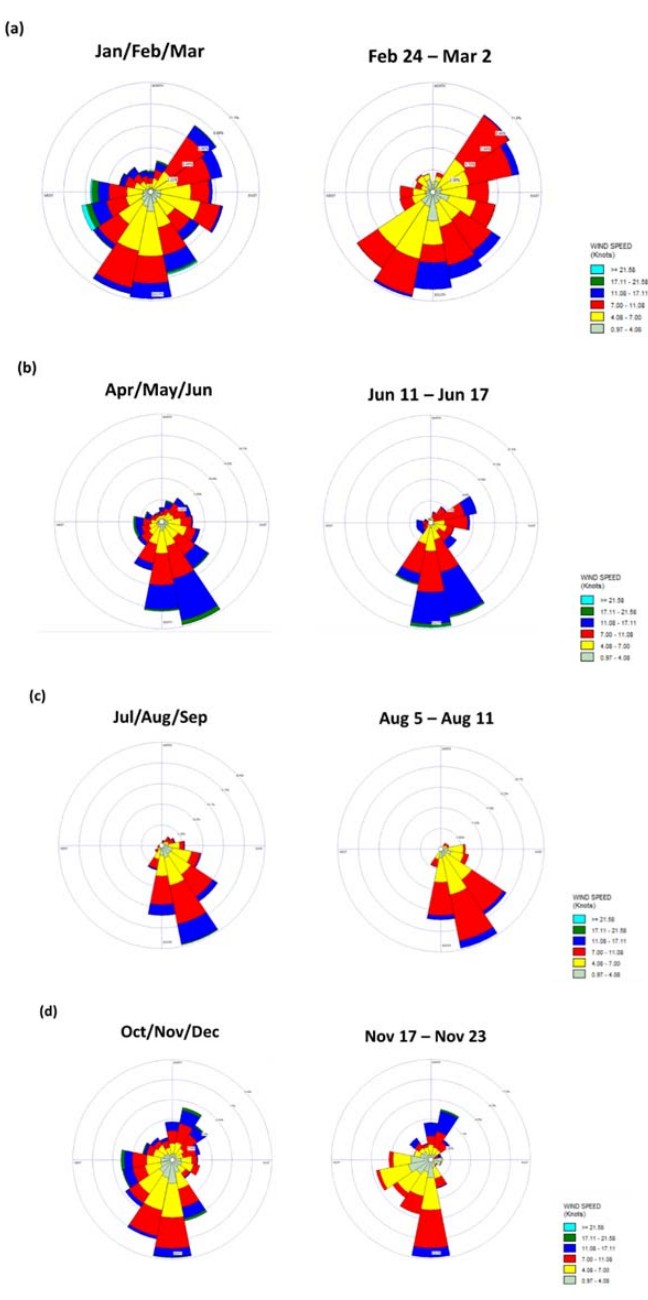




An oil and gas production site within the region was chosen for dispersion modeling. The site contained a well and associated

equipment, oil and water storage tanks, a vapor recovery unit, and a flare. An image of the site and potential sampling locations

are shown in Figure 3. Concentrations generated by emission rates of 10 kg/hr, from each of the sources, were used to estimate

observed concentrations at potential sensor sites. Emissions from the tanks and the vapor recovery unit were assumed to be at

5.5 m agl; emissions from the flare were assumed to be at 10 m agl.

**Figure 3. Representative site used to establish concentration thresholds for methane sensing systems; dispersion**

**modeling of emissions from the well and associated equipment, tanks, a vapor-recovery unit and a flare, were**

**estimated at potential sensor locations represented by red dots.**

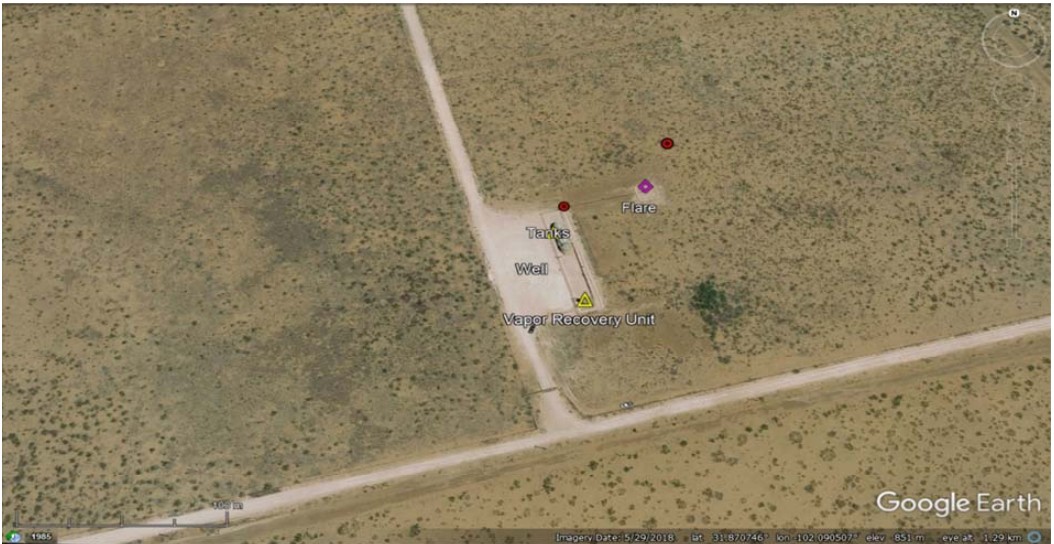

*Source* : "Representative site used to establish concentration thresholds for methane sensing systems" 31.870746° N and
102.090507° W. © **Google Earth**. May 29, 2018. December 15, 2021.

The CALPUFF (v7.2.1_L150618) dispersion model was used to predict concentrations of methane at the potential sampling

sites. CALPUFF is a non-steady state, Lagrangian puff modeling system (Exponent, 2014). The three-dimensional

meteorological fields used to drive the model were derived from the output of NOAA's (National Oceanic and Atmospheric

Administration) High Resolution Rapid Refresh (HRRR) atmospheric model (Benjamin et al., 2016). The model runs hourly

with a spatial resolution of 3km. The dispersion model was run at one minute temporal resolution with meteorological data

interpolated between the hourly outputs of the HRRR model.




### 2.2 Sensor testing

#### 2.2.1 Sensors tested

The four sensor systems which were evaluated in detail are described in Table 1. Brief descriptions of the sensors are provided below. More details of the operational principles of these and other types of sensors have been summarized by Wiley and

Moore (2021). Additional descriptions are available in the project final report (University of Texas, 2021). Three additional sensor systems that were deployed but that did not provide a sufficient quantity of data to allow detailed performance analyses are also described in the project final report (University of Texas, 2021).

**Table 1. Sensors**

|  | Quanta 3 | Aeris | Scientific Aviation | Project Canary |
|---|---|---|---|---|
| **Footprint/Space** | 30 x 30 inches | 6-9 ft$^2$ | 1 ft$^2$ | 2 - 9 ft$^2$ |
| **Temporal resolution** | 1 sec. | 0.5 - 1.0 sec. | 0.2 sec. | 1 sec. samples averaged to 1 min. |
| **Detector** | Near infrared laser absorption spectroscopy sensor based on tunable diode laser spectroscopy | Direct absorption mid-IR spectroscopy in the 3 micron region; able to quantify methane and ethane | Metal oxide sensor | Laser-based spectroscopy |
| **Range** | Ambient (2 ppm) to 500 ppm (optimized for high sensitivity) | 1ppb level to % concentration levels. | 500 ppb over background to 5000 ppm | Methane: 0.2ppm - 100ppm |
| **Sensitivity/Accuracy** | Field precision is better than 20 ppb; Accuracy is rated at ~20% at background concentrations (2 ppmv) | Accuracy: 1-2ppb drift long-term | precision: ± 60 ppb for a 1-minute average accuracy: ±1 ppm + 15% | Methane only : Precision: 0.001ppm; Accuracy: ~0.2ppm (~2 σ) |


*Aeris:* The Aeris Technologies sensor is a tunable mid-infrared laser-based gas analyzer with a sample flow rate of approximately 0.3-0.5 lpm. The system measures methane, ethane and water concentrations. The instrument reports continuously the gas dry mole fractions at a frequency of 1 Hz and data are uploaded to a server every other minute.

*Canary:* The Project Canary system that was tested used two sensors, one that measures methane and one that measures all

C3+ (compounds containing 3 or more carbons) using laser-based spectroscopy (methane) and photoionization (C3+) technologies. The system makes one measurement per second and reports minute averaged values via cellular modem to a cloud platform.

*Quanta3:* The Quanta3 sensor measures methane using tunable laser diode spectroscopy technology. The system makes one measurement every second and reports data at the same frequency via 4G/LT cellular modem to a cloud platform.





*Scientific Aviation:* The Scientific Aviation unit is a multi-gas sensor using metal oxide semiconductor technology. The system makes 5 voltage measurements per second and averages the 300 voltage measurements from each minute to report one voltage measurement per minute via LTE cellular modem to a cloud server. Voltage readings are converted to concentration measurements on the server.

**2.2.2    Site description**

The sampling was performed adjacent to an oil and gas production site with compressors, tanks, and a flare. The sensor inter-comparison site, with sensors installed, is shown in Figure 4. Sensors were aligned perpendicular to the prevailing wind direction on the downwind side of the site. Sensors were mounted in such a manner to be sampling the air at 2 – 3 meters above ground level, with an average of 4 meter center to center separation (depending on sensor footprint), each with clear

line of sight to the nearby sources. The exact sampling height was selected to avoid any conflicts with ancillary equipment such as solar panels, service boxes, and antennae.

**Figure 4.** Sampling site with sensors installed

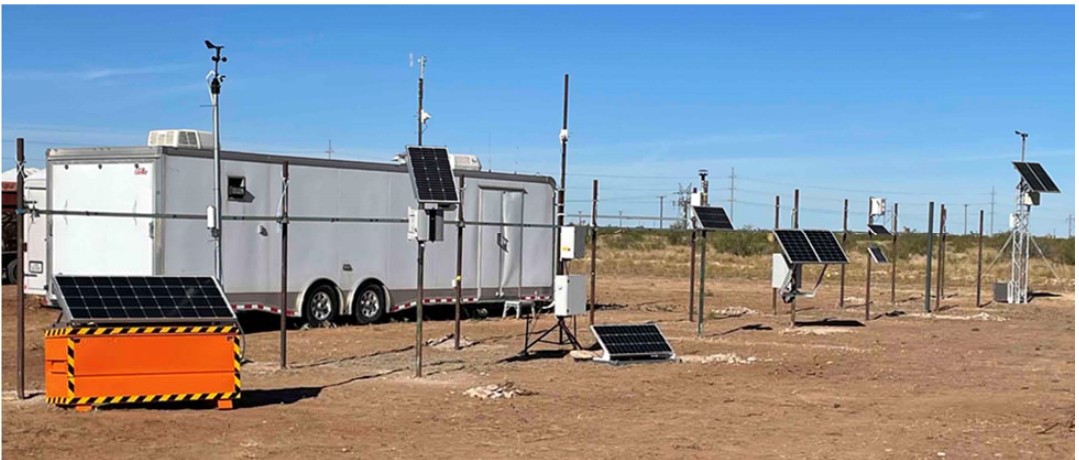

**2.2.3    QC-TILDAS measurements**

A quantum cascade tunable infrared laser differential absorption spectrometer (QC-TILDAS) (Nelson, et al., 2004; Roscioli, et al., 2015) was deployed to measure ambient methane concentrations and to confirm certified gas challenge compositions. The QC-TILDAS provides 1 Hz measurements of methane concentrations at 1 ppb precision with minimal interferences. The spectrometer was installed in the site trailer (see Figure 4) within an environmental chamber that provided a constant temperature environment for the instrument. A constant temperature bath circulated water through the QC-TILDAS to control



the instrument's laser within +/- 1°C.  Calibration gases were used to check the initial performance of the system.  Details of
        the QC-TILDAS calibration are described in the technical report of the study (University of Texas, 2021).

        The QC-TILDAS was programmed to sequentially draw its air sample from sampling poles located within 1-2 m of, and at
        the height of, each of the sensors being tested.   Each sensor location had a dedicated QC-TILDAS sampling line and ambient
air was continuously drawn through the lines by a vacuum pump.  All of the individual sampling lines, with continuous sample
        flow, were delivered to a multi-port sampling valve located within the sampling trailer.  The valve was placed as close to the
        QC-TILDAS instrument as possible enabling the shortest possible sample line between the valve and the QC-TILDAS
        instrument.  This minimized the time for sample line clearing associated with switching between ports, and maximized the
        sampling frequency at individual ports. The timing of the sequencing was designed to rapidly cycle through the sample
locations, while still allowing for a rapid clearing of the line between the multiport valve used to control the sequencing and
        the instrument.  A 12-minute period at each sample pole was selected to balance the small amount of data loss as sample lines
        were switched, with the practical need to simplify later matching up QC-TILDAS measurements with the sensor
        measurements. The QC-TILDAS data logger recorded the time for switching from one sample line to the next. A minimum of
        10 seconds of sampling was discarded at the start of each comparison to allow the air arriving at the next sensor to reach the
QC-TILDAS analyzer and for the gas in the sample chamber of the instrument to be exchanged.

### 2.2.4     Sampling period

Sampling was conducted from October, 2020 through June 22, 2021.  Sensors operated continuously throughout this period,
and data capture rates were determined.  Multiple extreme weather events occurred during the sampling period, including
Winter Storms Uri and Viola that disrupted local power and other infrastructure during February 2021.

### 2.2.5     Certified gas challenges

Multiple certified gas challenges were conducted during the nine month field deployment. Challenges delivered gas with
known methane, ethane and propane concentrations to sensor inlets, using interfaces specifically designed for each sensor.
The interfaces are described in the project final report (University of Texas, 2021). Gas concentrations delivered to each sensor
were varied in these challenges at intervals defined by the temporal resolution of the instrument being tested.  The challenge
gas cylinders' compositions were similar to the typical produced gas composition in the Permian Basin, (75 mole percent
methane, 15 mole percent ethane, and 10 percent mole percent propane). Methane mixing ratios of the four certified challenge
gases were 2.1 ppm, 2.2 ppm, 10 ppm, and 100 ppm (±10%, as certified by the gas provider).  The challenge gas concentration
of methane used at specific times during the tests were known only by the inter-comparison team, not the sensor operators,
making the challenge single blind. While the challenge gas was being applied, a slip stream of challenge gas was sent directly
to the QC-TILDAS instrument so that the response of the sensor and the response of the QC-TILDAS instrument could be
directly compared.






### 2.2.6 Ambient concentration comparisons with QC-TILDAS instrument

The measurements of the sensors were also compared to the measurements of the QC-TILDAS in the time periods between certified gas challenges. In performing the comparison for each sensor, the QC-TILDAS data were screened so that only the data sampled from the sampling location directly adjacent to the sensor being evaluated were considered. This means, even

for identical sampling periods, the QC-TILDAS time series was different for each sensor comparison. Before quantitatively evaluating sensor performance, the QC-TILDAS data were converted from 1-second to 1-minute resolution to minimize the impact of small time mismatches.

### 3 Results and Discussion

**3.1 Representative time series**

Figure 5 shows a typical week of data collected by the QC-TILDAS instrument. Measurements are reported using two vertical scales. The upper time series in the Figure shows concentrations capped at 100 ppm, the maximum concentration used in the QC-TILDAS calibration. The lower time series in the Figure shows concentrations capped at 10 ppm. As described in Section 3.2, it is expected that detections of concentrations in the 3-10 ppm region will be the basis for detecting emissions in sensor

networks. Concentrations in excess of 10 ppm were relatively infrequent.

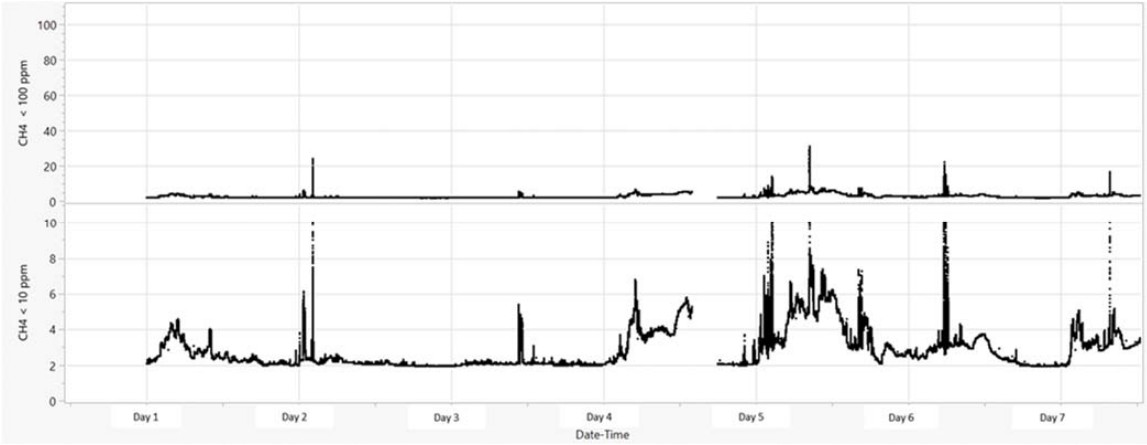

**Figure 5. QC-TILDAS measurements of ambient methane concentrations for one week-long period during the sensor**

**testing; measurements are shown with two vertical scales**





### 3.2 Sensor performance criteria

Dispersion modelling was used to predict concentration enhancements associated with emission rates of 10 kg/hr from the tanks, vapor recovery units and flare, with one minute temporal resolution, at the potential sensor locations shown in Figure 3. The meteorological data used in the dispersion model was representative of conditions in each of four seasons, as described in the methodology section. For the emissions of 10 kg/hr from the tanks, the sensor location at the northeastern corner of the well pad was predicted to have enhancements of more than 2 ppm in the mixing ratio of methane in 20% of predictions. Mixing

ratio enhancements of 1 ppm and 500 ppb were observed in 26% and 32% of the predictions, respectively. For the vapor recovery unit emissions, the same sensor location had predicted mixing ratio enhancements of at least 2 ppm, 1 ppm and 500 ppb in 9%, 22% and 30% of the predictions. For these two sources, a sensor able to detect mixing ratio enhancements of 500 ppb to 1 ppm would enable rapid detection of emission rates in the range of 5-10 kg/hr. For the flare, the northeast corner of the well pad is not as frequently downwind and for flare emissions the location had predicted mixing ratio enhancements of at

least 2 ppm, 1 ppm and 500 ppb in 0%, 2% and 4% of the predictions. Different locations, northeast of the flare are more frequently downwind of the flare and had predicted mixing ratio enhancements of at least 2 ppm, 1 ppm and 500 ppb in 0%, 3% and 6% of the predictions, suggesting that the relatively high release height of the flare emissions will make detection more challenging than detection from sources with lower release heights, however, reasonably frequent detections are still possible. Overall, dispersion modeling analyses suggest that sensors able to detect mixing ratio enhancements of 500 ppb to 1 ppm at a

one minute resolution should be able to detect emissions of 5-10 kg per hour relatively quickly at many oil and gas production sites in the Permian Basin.

### 3.3 Certified Gas Challenges

Figure 6 shows the response of the QC-TILDAS to multiple 100 ppm gas challenges. Each challenge lasted several minutes

and was separated from the previous challenge by one to several minutes. The data show that the QC-TILDAS instrument responded rapidly to the challenge gases, typically reaching steady state concentrations within ~10-15 seconds; steady state concentrations were stable and reproducible. The performance metrics used in evaluating the four sensors relied on some of these QC-TILDAS challenge gas response data. Specifically, the metrics used in the sensor evaluation were:

- Rise Speed – estimated time duration for the measured concentration to reach 2/3 of maximum concentration recorded

by the sensor being tested during the challenge. This was, for example, typically 60 - 66 ppm for the 100 ppm gas, and was more variable for the lower concentration gases due to variable background concentrations. The rise time includes a lag associated with the volume of the interfaces used to deliver challenge gases to the sensors being tested. For the QC-TILDAS, this was 10-15 seconds, so response times below this level are not expected.

- Decline Speed – estimated time duration for the measured concentration to reach 2/3 of the difference between the

response to the challenge gas and the ambient background concentration, once the challenge gas feed was removed.





The decline time includes a lag associated with any residual certified gas in the region of the sample inlet once the interface used to deliver challenge gases is removed. For the QC-TILDAS, this was 10-15 seconds, so response times below this level are not expected.

- Percent of Target Concentration – the maximum concentration reached by the sensor compared to the coincident QC-
TILDAS mean concentration during the challenge.
- Integrated response Relative to QC-TILDAS – Concentration responses were integrated over the duration of the test for the QC-TILDAS and the sensor. For example, a 100 ppm challenge gas applied to the QC-TILDAS for 4 minutes produces 240 seconds at 100 ppm, or 24,000 ppm-secs, and a 4-minute 10 ppm gas applied to the TILDAS produces 240 seconds at 10 ppm, or 2,400 ppm-secs.


Table 2 reports the results of the challenge gas tests, using these metrics. The Table summarizes the results of challenging the four sensors with 10 ppm and 100 ppm calibration gases over the study period.

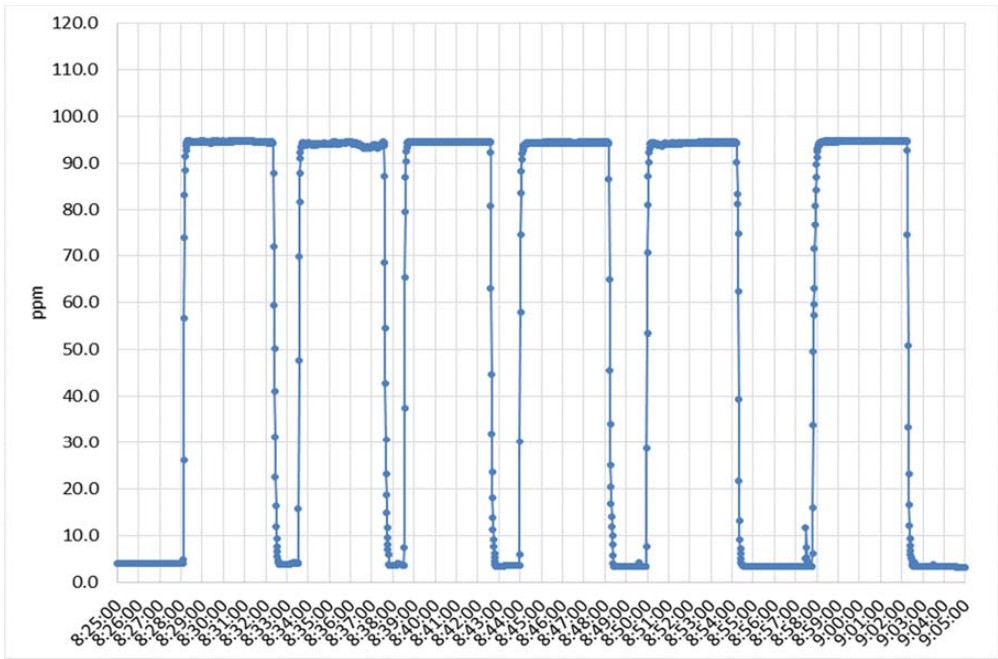

**Figure 6. Response of QC-TILDAS instrument to repeated gas challenges; data for 100 ppm are shown**



**Table 2. Summary of performance over all the challenge gas tests**

| Sensor | CH₄ gas concentration | Number of comparisons | Rise Time Average | Decline Time Average | Integration % Average and 95% C.I. | Peak % Average and 95% C.I. |
|---|---|---|---|---|---|---|
| Scientific Aviation | 10 ppm | 8 | ~ 1min | 1-2 min | 82.8 ± 30.5 | 98.3 ± 37.4 |
| | 100 ppm | 8 | 1-2 min | 1-2 min | 94.0 ± 29.7 | 103.6 ± 27.2 |
| Aeris | 10 ppm | 6 | 8 sec | 20 sec | 99.7 ± 3.6 | 103.7 ± 2.4 |
| | 100 ppm | 7 | 8 sec | 17 sec | 91.3 ± 6.7 | 100.9 ± 1.3 |
| Canary | 10 ppm | 8 | ~ 2min | 1-2 min | 61.8 ± 13.5 | 74.6 ± 12.2 |
| | 100 ppm | 7 | ~ 2min | 1-2 min | 52.7 ± 13.0 | 73.7 ± 15.1 |
| Quanta3 | 10 ppm | 8 | 10 sec | 31 sec | 91.8 ± 8.2 | 98.0 ± 4.7 |
| | 100 ppm | 8 | 13 sec | 24 sec | 92.5 ± 9.3 | 95.6 ± 7.0 |


### 3.4 Ambient concentration comparisons with QC-TILDAS instrument

Figure 7 shows comparisons of one-minute averaged QC-TILDAS signals and the signals from each of the individual sensors for a representative week of sampling. Note that because the comparisons are only done for periods when the QC-TILDAS is sampling directly adjacent to the sensor being evaluated, the QC-TILDAS signals differ for the individual sensor comparisons.

Since dispersion modelling reported in Section 3.2 suggested that the ability to detect mixing ratio enhancements of ~1 ppm would be required to detect emission rates in the range of 5-10 kg/hr, distributions of observed methane mixing ratios observed by the QC-TILDAS and the four sensors were compared. Representative results for one month of data are shown in Figure 8. In addition, best fits of a linear relationship between the QC-TILDAS measurements and the sensor measurements were determined for each month of sampling, and are shown in Table 3. The Aeris, Quanta3 and Canary sensors distributions of

observed concentrations show good agreement with the QC-TILDAS instrument. Slopes of the best linear fit of the sensor concentration to the QC-TILDAS concentration, which had its sample inlet 1-2 meters from each of the sensors being tested were generally >0.6, for concentrations averaged over one minute. Linear correlation coefficients ($R^2$) of these two concentration measurements were generally >0.6. The Scientific Aviation sensor showed baseline bias and this is evident in both the distributions of observed concentrations and the linear correlation.


Additional analyses were carried out to compare sensor responses, relative to a baseline, to the QC-TILDAS response, relative to a baseline. The baseline chosen was the minimum daily concentration recorded by the sensor, although other baselines could also be selected (e.g., rolling minimum concentration over the hour before the measurement). Plots of minimum daily concentrations, for the entire study period, are shown for the QC-TILDAS, Aeris, Canary, Quanta3 and Scientific Aviation



sensors in Figure 9. When the baseline corrected Scientific Aviation signal was compared to the baseline corrected QC-
TILDAS signal, the bias in the Scientific Aviation sensor was significantly reduced, as shown in Figure 10.





a.

b.


c.

d.

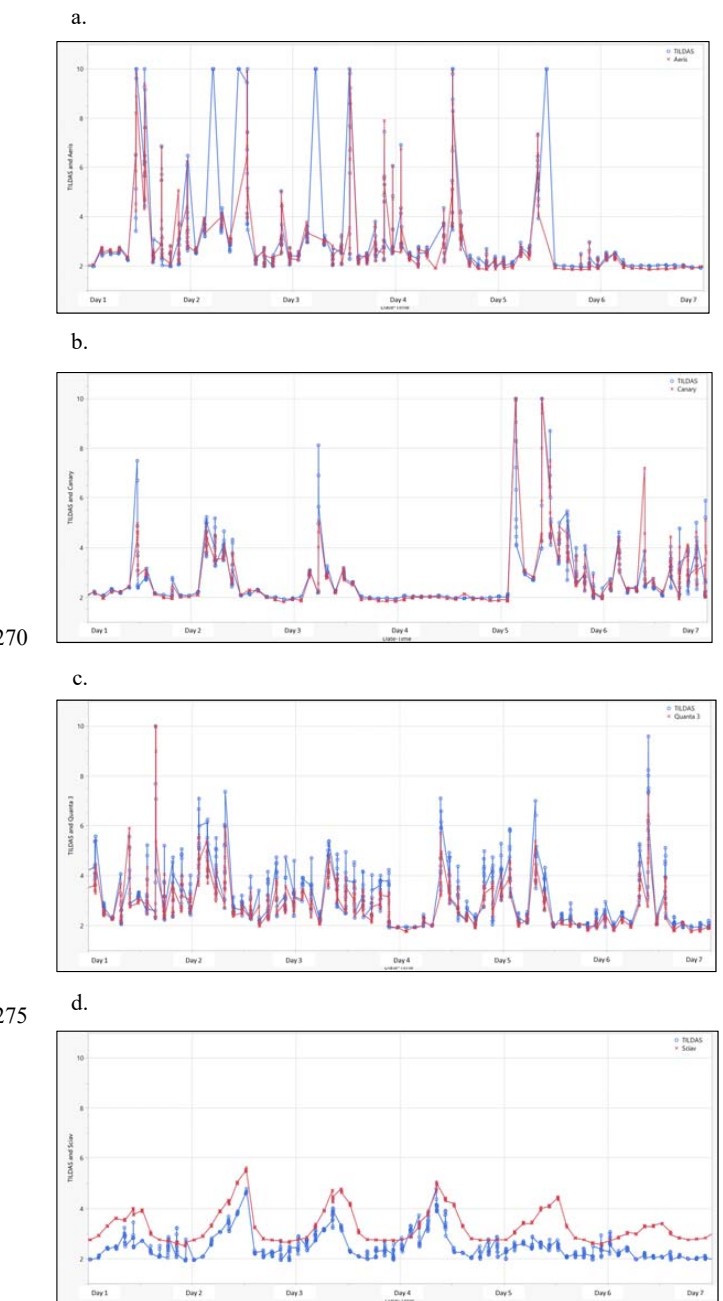

**Figure 7. Comparisons between 1-minute averaged QC-TILDAS and (a) Aeris (b) Canary, (c) Quanta3 and (d) Scientific Aviation sensors for a representative week of data**


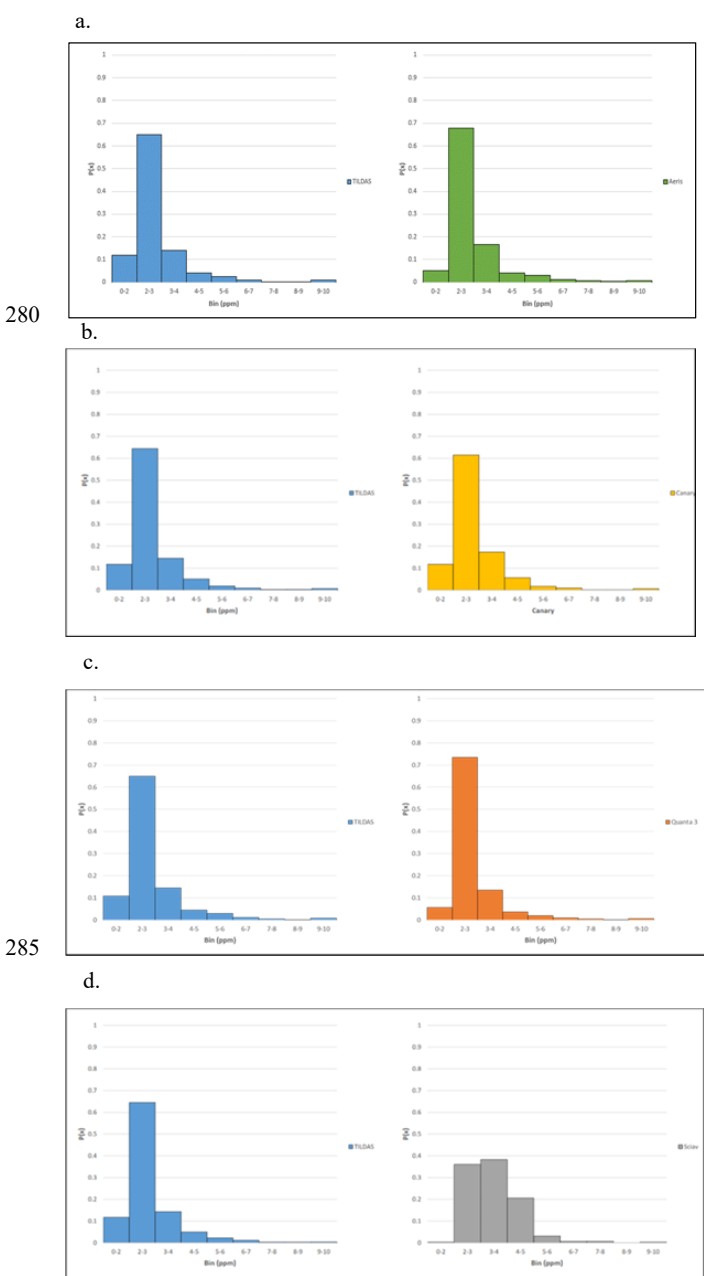



**Figure 8. Comparisons between distributions of 1-minute averaged concentrations measured by QC-TILDAS and (a)**
**Aeris (b) Canary, (c) Quanta3 and (d) Scientific Aviation sensors for a representative month of data.**






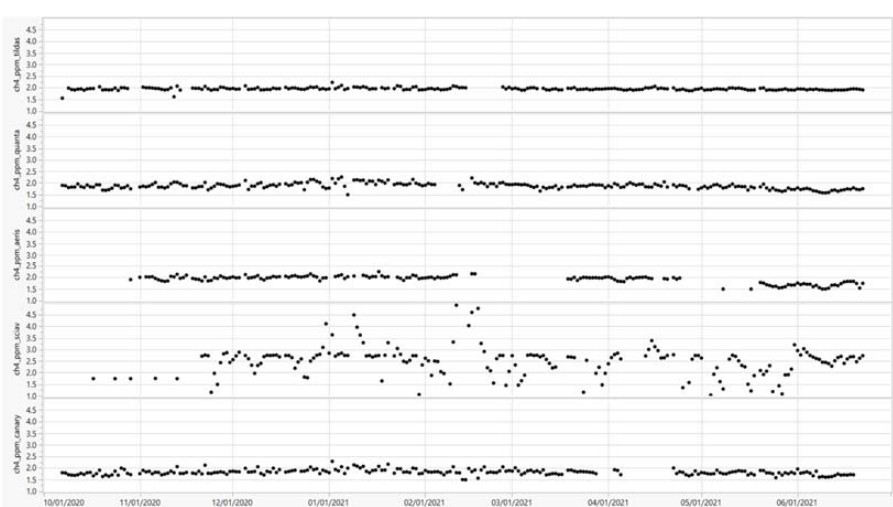

**Figure 9.  Minimum daily concentrations recorded by the QC-TILDAS and the Aeris, Canary, Quanta3 and Scientific Aviation sensors.**



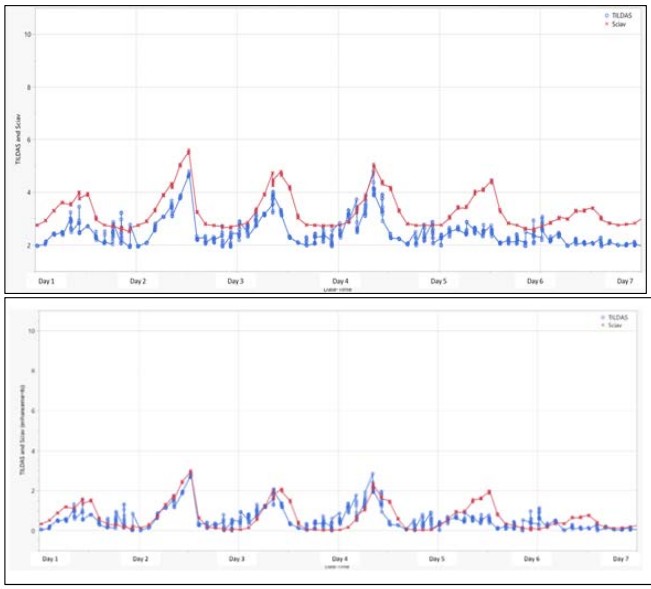


**Figure 10.  Comparisons between 1-minute averaged QC-TILDAS and Scientific Aviation sensor with and without background correction**






**Table 3. Monthly summaries data reporting as scatter plots and regression fits to 1-min averaged QC-TILDAS and sensor data**

|  | Aeris | Canary | Quanta 3 | Scientific Aviation |
|---|---|---|---|---|
|  | Slope of Sensor Measurement (y) vs. QC-TILDAS Measurement (x) [$R^2$ of correlation] | | | |
| October | N/A | 0.95[0.92] | 0.87[0.96] | 0.14[0.08] |
| November | 0.94[0.87] | 0.91[0.90] | 0.72[0.92] | 0.38[0.40] |
| December | 0.85[0.85] | 0.55[0.61] | 0.83[0.93] | 0.57[0.46] |
| January | 0.90[0.87] | 0.99[0.85] | 0.70[0.73] | 0.83[0.67] |
| February* | 0.96[0.94] | 0.97[0.89] | 0.91[0.95] | 0.41[0.11] |
| March | 0.86[0.71] | 0.76[0.83] | 0.60[0.89] | 0.52[0.46] |
| April | 0.94[0.87] | 0.82[0.92] | 0.62[0.82] | 0.72[0.71] |
| May | 0.26[0.34] | 0.95[0.79] | 0.70[0.90] | 0.49[0.49] |
| June | 0.75[0.58] | 0.72[0.72] | 0.77[0.93] | 1.0[0.74] |
| Full data set | 0.77[0.68] | 0.79[0.78] | 0.67[0.85] | 0.75[0.56] |

*February 2021 a partial month owing to bad weather

**3.5 Data capture**

All four sensor systems had greater than 80% data completeness, as reported in Table 4. The data in Table 4 exclude the period from February 15-25, 2021, when winter storms Uri and Violet caused widespread failure of local infrastructure. Missing data was sometimes caused by interruption of cellular service, or delays in having an operator visit the site to reboot the sensor. One sensor system (Scientific Aviation) operated throughout the period of the winter storm and had 100% data completeness

during the inter-comparison testing.

**Table 4. Data Completeness (excluding Feb. 15 -25, 2021)**

| Sensor System | Period of Operation | Data completeness |
|---|---|---|
| QC-TILDAS | 10/7/2020 to 6/22/2021 | 92.5% |
| Aeris | 10/28/2020 to 6/22/2021 | 84.4% |
| Canary | 10/1/2020 to 6/19/2021 | 91.5% |
| Quanta 3 | 10/7/2020 to 6/22/2021 | 95.9% |
| Scientific Aviation | 10/1/2020 to 6/22/2021 | 100% |


**4 Conclusions**

Four solar powered methane sensing systems demonstrated the ability to detect methane concentration enhancements in the range of 500 ppb-1 ppm, at one-minute time resolution, in extended field testing in an oil and gas production region in west Texas. Dispersion modeling indicates that these concentration enhancements are consistent with methane emission rates of 5-



10 kg/hr, if the sensors are located within approximately 50-100 m of the sources. All four sensors had data capture rates that exceeded 80% during 9 months of operation, despite severe weather conditions and extended local electrical power losses. These results demonstrate that multiple commercially available sensing systems are suitable for long term methane emission monitoring in remote oil and gas production regions.

**Aknowledgements**

Support for this work was provided by AT&T, Environmental Defense Fund, ExxonMobil, and the Collaboratory to Advance Methane Science. Sensor providers donated their instrumentation and their data services.

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
