# Peer review of "Field inter-comparison of low-cost sensors for monitoring methane emissions from oil and gas production operations"

_Atmospheric Measurement Techniques, 2022_

## Author Comment (AC1)

**Responses to reviewer comments**

**amt-2022-24**

**Field inter-comparison of low-cost sensors for monitoring methane emissions from oil and gas production operations**

**Torres, et al.**

**Reviewer 1:**

Comment: Overall, this study strongly advances the important topic of near-source emissions detection for oil and gas applications. This is a high-quality study of significant scope, and the manuscript is generally well-written. This work should be of interest to the readers of this journal and a wider audience. In this reviewer's opinion, this manuscript should be published after considering revisions.

The authors conclude that the tested sensors demonstrate the ability to detect methane concentration enhancements in the range of 500 ppb to 1 ppm at 1-min time resolution. Coupled with impressive data completeness, the authors conclude that these systems are suitable for long term methane emissions monitoring at oil and gas sites. The analysis presented generally supports the detection performance statement for certain monitoring conditions, but the analysis could be significantly strengthened regarding the primary monitoring objective of detection of emission plumes. Currently, it is not clear that all sensors can detect rapidly changing concentrations indicative of near-field source emissions at the stated performance objective.

The modeled sensor performance criteria in Section 3.2 states that a sensor should be able to detect enhancements of 500 ppb to 1 ppm over background with 1 min time esolution. However, the analysis does not strongly examine sensor performance against these levels. Table 2, for example, shows gas challenges at 10 ppm and 100 ppm but data should be available at 2.2 PPM as well.

*Response: Data for the response at 2.1 and 2.2 ppm has been added to the manuscript; the added text and table is (blue font):*

Using these metrics, Table 2 reports the results of the challenge gas tests for the four sensors with 10 ppm and 100 ppm calibration gases over the study period. Table 3 reports the results for the 2.1 and 2.2 ppm calibration gases. For the 2.1 and 2.2 ppm challenges, the objective was to test the ability of sensors to discriminate between close concentrations near the background concentration. Table 3 reports the difference between responses to the 2.1 and 2.2 ppm challenge gases for each of the sensors.

**Table 3. Summary of responses to the 2.1 and 2.2 ppm challenge gas tests**

| Sensor | CH$_4$ gas concentration | Number of comparisons | Sensor mean response | Mean difference 2.1 vs 2.2 ppm |
|---|---|---|---|---|
| Scientific Aviation | 2.1 ppm | 8 | 5.44 | -0.819 ± 0.883 |
| Scientific Aviation | 2.2 ppm | 8 | 4.62 | |
| Aeris | 2.1 ppm | 6 | 2.04 | 0.106 ± 0.026 |
| Aeris | 2.2 ppm | 6 | 2.14 | |
| Canary | 2.1 ppm | 7 | 2.83 | 0.013 ± 0.167 |
| Canary | 2.2 ppm | 7 | 2.84 | |
| Quanta 3 | 2.1 ppm | 8 | 2.14 | 0.080 ± 0.118 |
| Quanta 3 | 2.2 ppm | 8 | 2.22 | |

Comment: Regarding comparisons to QC-TIDLAS, concentration enhancements observed by near-source sensors typically represent a superposition of slowly varying background signal and rapidly varying emission plume signal from the potential emission source under study. The performance criteria for source-proximate emission detection approaches should center on the sensor's ability to detect proximate emission plumes. A sensor's ability to track slow diurnal changes in methane with high accuracy is somewhat less important. This paper could be strengthened by adding a subset analysis focused on temporally sharp, multi-ppm enchantments likely representing plume signal from the adjacent site. For example, using QC-TIDLAS determined short term excursions (e.g. > 5 ppm), what percentage of these peaks were successfully detected by the sensors under study. This type of analysis will separate slowly varying background data from source-induced concentration enhancements (the primary application).

 As it stands, the ability of the sensors to track dynamic concentration changes indicative of near-field emission plumes is difficult to understand. For example, Figure 7(d) is illustrative of baseline offset but lacks the 5-ppm signal excursions for comparison to other cases in the same figure. Looking at Table 3, how much of the decorrelation in the slow MOX sensor is due to baseline drift and how much is due to insufficient temporal response to rapidly varying plume signal that is properly captured by QC-TIDLAS reference instrument?

*Response: The reviewer correctly identifies slow response as one of the reasons for the decorrelation between the metal oxide sensing system and the QC-TILDAS reference instrument. We have added text and figures highlighting this issue in the revised manuscript. Rapidly varying ambient concentrations make it difficult to quantitatively account for this lag with a simple delay in response, therefore we have added text describing the average concentrations recorded by the metal oxide sensor when the QC-TILDAS instrument was recording*

*measurements in various concentration measurements (unmatched in time). The revisions to the text are given below (blue font)*

In addition to the baseline correction, Figure 10 also suggests a delayed response for the Scientific Aviation sensor, relative to the QC-TILDAS sensor. Examples of this delay are shown in Figure 11. Rapidly varying ambient concentrations make it difficult to quantitatively account for this lag with a simple delay in response, however, the average concentration recorded by the Scientific Aviation sensor can be calculated for periods when the QC-TILDAS instrument was recording measurements in various concentration measurements. For example, for the 3,709 minutes when the QC-TILDAS instrument recorded nixing ratios greater than 20 ppm at the sampling site adjacent to the Scientific Aviation sensor (mean of 30.8 ppm), the Scientific Aviation sensor recorded a mean mixing ratio of 24.3 ppm. For the 13,927 minutes when the QC-TILDAS instrument recorded mixing ratios greater than 10 ppm (mean of 18.0 ppm), the Scientific Aviation sensor recorded a mean mixing ratio of 13.7 ppm. These results suggests that the Scientific Aviation sensor is generally detecting methane enhancements over background, but separately accounting the impacts of baseline drift and time lags is challenging.

[Figure]

**Figure 11. Examples of time lags between the QC-TILDAS and Scientific Aviation measurements**

---

## Author Comment (AC2)

amt-2022-24

**Field inter-comparison of low-cost sensors for monitoring methane emissions from oil and gas production operations**

**Torres, et al.**

**Reviewer 2**

Comment: This work attempts to address an important issue in relation to the large methane emissions from oil production, in which low-cost sensors (LCS) are evaluated as a possible alternative to expensive measurement instrumentation. However, the way the article is organized makes it difficult to follow, the objectives are vague, the methodology is not clearly outlined - which undermines the reproducibility of the results-, and the discussion of the findings and their potential impact is absent in the text. It is the consideration of this reviewer that this manuscript requires further maturation, therefore, major changes are suggested.

Three things seem to be crucial and require better description, further development, and deeper analysis: the dispersion model (and on which some of the subsequent efforts depend), the devices under evaluation (which must meet certain criteria to be considered fit for the purpose) and the last is the colocation experiment (which allows their evaluation).

Since the use of the Calpuff model is crucial for the definition of the assessment criteria as is presented here, is essential to describe the details of the modelling exercise. What configuration was used? What other inputs did it require? What are the most relevant uncertainties regarding the problem in question? What is the spatial and temporal distribution predicted by the model? Are there similar works in the literature, in which a dispersion model has contributed to defining this type of criteria? The Calpuff model has a 3 km spatial resolution, so if the model was run at this standard resolution, how were the results interpreted considering that the sources are located just a few meters from the sensors? Or is it perhaps that the code was modified to improve its resolution? Using only a few weeks' worths of weather may also contribute to limiting the scope of the results, and perhaps this deserves further attention. It is also not clear how meteorology of 1 min time res. was obtained (the work mentions that hourly data was interpolated, which seems wrong. See specific comments below).

*Response: We have added text describing more of the details of the Calpuff modeling. The added text is provided below (blue font):*

The CALPUFF (v7.2.1_L150618) dispersion model was used to predict concentrations of methane at the potential sampling sites. CALPUFF is a non-steady state, Lagrangian puff modeling system (Exponent, 2014). Calpuff requires emissions release characteristics, together with a representation of 3D meteorological conditions and geophysical characteristics. The geophysical datasets for the modeling domain were obtained from the National Elevation Dataset 2013 (~10 m resolution) and the National Land Cover Database 2017 (USGS, 2013, 2016). CalPuff has a set of pre-processors (CTGPRC, TERREL and MAKEGEO), to read the ingested datasets and assign appropriate values to user-specified locations. The three-dimensional meteorological fields used to drive Calpuff

were derived from the output of NOAA's (National Oceanic and Atmospheric Administration) High Resolution Rapid Refresh (HRRR) atmospheric model (Benjamin et al., 2016). HRRR has a spatial resolution of 3km and a temporal resolution of 1 h. The Mesoscale Model Interface Program (MMIFv3.4.1) was used to process the meteorological dataset from HRRR Model to CalPuff-supported format. (https://gaftp.epa.gov/Air/aqmg/SCRAM/models/related/mmif/MMIFv3.4.1_Users_Manual.pdf). The coarser dataset is then linearly interpolated by Calpuff to create a finer gridded dataset and predict concentrations at user-specified locations at a temporal resolution of 1 minute.

The observations of wind speed and wind directions at the nearest grid cell of the modeling domain were extracted from HRRR and compared to the observations at the monitoring site maintained by the Texas Commission on Environmental Quality (TCEQ) for the four representative meteorological weeks. The four meteorological weeks selected captured the variations in the wind speed and direction for the entire year reasonably well. Some changes in meteorological parameters that occur within an hour interval will not be captured accurately with the linear interpolation used in CalPuff. This behavior will result in some uncertainties in the concentration predictions, however, these fine time scale changes are unlikely to affect the conclusions regarding the precision required of sensors.

[References added:]

United States Geological Survey (USGS). National Elevation Dataset, URL: https://data.tnris.org/collection/e0ead9bd-0c01-4716-97e9-808ec330afd22013-01-01. Web. 2022-04-15.

United States Geological Survey (USGS). National Land Cover Database, 2016-12-31. URL: https://data.tnris.org/collection/89b4016e-d091-46f6-bd45-8d3bc154f1fc2016-12-31. Web. 2022-04-15.

Comment: As for the low-cost sensors (words that appear only once in the entire text), important issues are not considered by the authors. For example, interferences and drift that this cheaper hardware usually suffers from is something that needs to be considered. It would then be important to discuss the different techniques used by these devices and what are the potential problems that they may experience in relation to the measurement technique they use. Furthermore, if these systems have already been used in other scientific studies and/or evaluated, it may also be useful to understand other potential limitations. This discussion is absent in the manuscript and is essential to understand the potential uses and limitations in this oil industry scenario.

*Response: We have deleted "low-cost" from the title, noting that the inter-comparison we report also include data from the QC-TILDAS sensor which had a cost >US$150,000. We have also added additional text and Figures addressing the baseline drift and time lag associated with the metal oxide (Scientific Aviation) sensor, in response to comments from Reviewer 1. We do not repeat that response here. We have also added the following text to the revised manuscript (blue font):*

The nature of the metal oxide sensor in the Scientific Aviation sensor makes it vulnerable to interference by water content in the atmosphere. This may be correctable. A weak correlation with relative humidity (RH) was observed in the data ($\rho$=0.32).

Comment: The colocation experiment used to determine performance also requires a deeper description and analysis. This is perhaps would be the richest aspect of the manuscript and needs

to be exploited. Something that catches my attention is that the Tildas is said to be operated continuously, however, the results of the comparisons seem to leave out some elements that could contribute to understanding the usefulness of the LCS in this scenario. The figures showed (and almost no discussion) seem to represent only bits of the colocation study (supposedly 9 months long). Although direct comparisons between sensor "x" and the reference instrument have been intermittent in those 9 months, it would be important to show and analyze the results throughout the entire period and if there are changes in performance (which is common in LCS) try to identify the possible causes or at least the observable impacts. Furthermore, if the reference instrument is continuously present at the site, it could also be used to test the initially assumed capabilities of the dispersion model and redefine (or not) the evaluation criteria. Complimentary to this certified gas challenges are used to compare LCS and Tildas instrument response. However, a final report (University of Texas, 2021) is cited repeatedly, but when this reviewer accessed it almost no details are shown (in contrast to what is said in the manuscript). This in other circumstances might not be relevant, but as the text presents it as an important source of information and on which this work seems to rest, it would be very important to be better describe how the experiment configuration was.

*Response: The reviewer raises several related points in this comment:*

*Intermittency in reporting: A single QC-TILDAS instrument sampled sequentially at locations adjacent to each of the sensors being tested, as described in the manuscript. This means that each co-located comparison has an intermittency.*

*Analysis of nine months of data: Table 4 in the revised manuscript [Table 3 in the original submission) summarizes 9 months of comparisons between the TILDAS instrument and the four sensors on a month by month basis to allow temporal changes, if any, to be shown. Although there are month to month fluctuations, overall trends are very weak.*

*Final report cited in manuscript: This report (~100 pages) was submitted as supporting information with the preprint and will be posted on the University of Texas web site after there is a decision on the manuscript.*

Comment: As a final consideration, I think that the work also needs to be improved regarding how it is structured and how the results are presented and analyzed. The figures presented are sometimes unclear and little explanatory, which hinders the reading quite a bit. The quality of the figures and tables must be improved throughout the text, as well as enriching the text with the analysis of the graphic results. The discussion and conclusions, almost absent in the manuscript, need to be greatly improved.

*Response: The purpose of this work is to report on the performance of a variety of methane sensing systems in oil and gas field operations. Sensors differed in their performance and these differences are described in the manuscript. Conclusions regarding which sensors are most appropriate will depend on the application. For example, ample space is available on production pad, allowing sensors to be placed very close to potential sources, sensors able to detect methane mixing ratio enhancements of 1 ppm or mere may be effective. In contrast, if pad space or other constraints require that sensors be placed on an adjacent pad, much more precise*

*sensors may be a more effective choice. Other work by our research team has been published describing the design of sensor networks and has been cited in the revised manuscript. Text added to the conclusions is shown below (blue font).*

Selecting among the four sensors evaluated in detail in this work will depend on the application. For example, in an application where ample space is available on a production pad, allowing sensors to be placed very close to potential sources, sensors able to detect methane mixing ratio enhancements of 1 ppm or mere may be effective. In contrast, if pad space or other constraints require that sensors be placed on an adjacent pad, much more precise sensors may be a more effective choice. The design of methane sensor networks, considering variations in sensor performance is reported by Chen, et al. (2022).

Chen, Q., Modi, M., McGaughey, G.M., Kimura, Y. McDonald-Buller, E.C., Allen, D.T., Simulated methane emission detection capabilities of continuous monitoring networks in oil and gas production regions, Atmosphere, 13(4), 510, doi: 10.3390/atmos13040510 (2022).

Specific comments

Title: the low-cost sensors are not mentioned elsewhere
*Response: "Low-cost" has been deleted from the title.*

Abstract: It is not clear what the contribution of the work is. The results highlighted here are somewhat sparse. Something that catches my attention is that the data capture is detailed, but results that could be more relevant are not considered.
*Response: We have expanded the conclusions, as noted above.*

Introduction: Although the use of LCS may be relevant, the motivation behind this study is not clear. This may be related to the fact that the reviewed literature is scarce. Regarding the potential of using dispersion models as an independent source of information, it is not reviewed either, which seems to have an impact on the methodology.
*Response: We have added to the citations of methane sensing of oil and gas production regions that was in the original text a reference to a recent review of methane emission identification, detection, quantification, and measurement methods.*

Line 31-34: it mentions the interest in using networks of sensors, which seems to be outside the scope of this study more focused on the evaluation of independent instruments (and not as part of a network).
*Response: The purpose of this work is to report on the performance of a variety of methane sensing systems in oil and gas field operations. As noted above, other work by our research on the design of sensor networks has been cited in the revised manuscript.*

Line 23-24: citation of own works seems somewhat excessive, even more so if one considers that the literature review referred to the topic that the paper tries to address is scarce. Here are some papers that could be considered in this matter:

https://www.sciencedirect.com/science/article/pii/S0048969721012614

https://www.sciencedirect.com/science/article/pii/S1352231020301771

https://www.sciencedirect.com/science/article/pii/S1352231018306241

*Response: We thank the reviewer for these suggested citations, which are generally relevant to the analysis of low cost sensors. We have added to the citations of methane sensing of oil and gas production regions that was in the original text a reference to a recent review of methane emission identification, detection, quantification, and measurement methods.*

Line 55: "The emission rate (5-10 kg/hr detectable at a distance of 50-100 m) and temporal resolution requirements (one-minute resolution) for sensors were converted into requirements for precision using dispersion modelling." This requires a more detailed explanation. What are the logical steps?

*Response: We have added more detailed description of the dispersion modeling and a citation to how sensor information can be used in the design of networks.*

line 59-66 (plus figures): "Results and discussion" seems to be a more appropriate place for this. It is also important to include the location of the mentioned station. How far is it from the site? Is it representative of the study site? What are the risks and uncertainties associated with considering only 4 one-week periods?

*Response: We have noted that the distance to the study site was approximately 20 km. Figures 1 and 2 compare the annual meteorological data to the shorter episodes.*

Figure 1 and 2 could be merged

*Response: Figures 1 and 2 could be merged, but separating them makes the distinction between the periods reported in the Figures more prominent, so they have not been merged.*

line 78: were the emissions modelled simultaneously or each emission source separately? constant or variable over time?

*Response: In the dispersion modeling, emissions were assumed to be constant. It is important to note that in this work the dispersion modeling was used only to establish general guidelines for sensor performance.*

Figure 3: the details are described above but also below the fig (it should be only below). The quality of the image needs to be improved. I suggest combining it with figure 4 or be sent to the supplementary. They both add very little info to the manuscript. In the text that accompanies the figure, potential locations for the sensors are discussed. What were the criteria for deciding the final location? What are the distances to the sources?

*Response: Since the reviewer asked for more detailed information about the dispersion modeling and expressed frustration about finding information in Supporting Information, we have left these Figures in the revised text.*

Line 89: It is suggested to create a new section to describe the model.
*Response: All of this belongs under the topic of establishing performance criteria. We have not created a new section.*

line 93: it says that the meteorology of 1-hour resolution was interpolated to one minute. It is not clear from the manuscript how this was done.
*Response: Additional details have been added regarding the dispersion modeling methods, as described above.*

Line 99: check the two opening sentences, they seem to say similar things.
*Response: The second sentence has been deleted.*

Table 1. The text is not explanatory. This table is not discussed/analyzed. In that case, it is suggested to be sent to the supplementary. Regarding the information contained, it is heterogeneous and due to the diversity of units used, comparisons cannot be made on the potential use of these instruments. Also, in the description below (line 107-119) some bits are repeated so it is suggested to harmonize this.
*Response: Since the primary focus of this manuscript is the comparison of sensors, we believe that a Table summarizing the characteristics of the sensors is important to retain in the manuscript.*

Line 122: The description of the site is probably better if it is included at the beginning of the methodology
*Response: We have retained the ordering, as we believe it is preferable.*

Line 132 says that the Tildas was used to confirm the certified gas. How was this instrument calibrated? Does it suffer from specific interferences?
*Response: The TILDAS and the calibration procedures are described in detail in the citations (Nelson, et al., 2004; Roscioli, et al., 2015, University of Texas, 2021)*

Line 147-150: Is this for the entire 9-month period? How often was this done?
*Response: We have added that the frequency of challenges was monthly; the number of challenges is also reported in Table 2.*

line 154-156: this seems to fit better at the beginning of the methodology.
*Response: We have retained the ordering, as we believe it is preferable.*

line 159: This part of the methodological design is important, and its description should be improved. Including a fuller explanation of how these experiments were performed would help replicate these experiments. Even including diagrams and schematics would be of great help.
*Response: These details were included in the Supporting Information (Final report) that was submitted along with the preprint. This final report will also be available at a University of Texas website once the manuscript is published.*

Figure 5: improved quality is needed (labels and contrast). Please add the dates and show the 9

months of colocation. Descriptive statistics would also offer great inside.
*Response:  This Figure was intended to illustrate representative variability in atmospgeric concentrations.  It is not clear what descriptive statistics would be useful.*

Line 195: the title seems to be wrong. Perhaps the model results could be included here?
*Response:  We believe the title "sensor performance criteria" is correct; the dispersion modeling results used to generally define some of the performance criteria are included in this section.*

Lines 220 to 235: the metrics' names used to assess the performance of the sensors (Rise Speed, Decline Speed, Percent of Target Concentration, and Integrated response Relative to QC-TILDAS) differ from those defined in Table 2. I suggest using the same names.
*Response:  We have used more complete, descriptive titles, too long to be used in Table column headings, in the main text.*

Figure 3. Improve the quality of the figure (labels). I suggest including the sensor's response to the challenges plots.
*Response: Our versions of the Figures appear to be of good resolution; if this is an issue in Figure conversion, we will address this with the publisher.*

Table 2 shows the concentrations used in the challenges (10 and 100 ppm), which exclude the expected concentrations according to the model (3-4 ppm). Furthermore, it is not clear when these experiments were performed. Were they one after the other? At different times of the year? At the beginning of the lifespan of the sensors? Did this behaviour change in time? How?
*Response:  We added the frequency of testing to the text (see above) and we have added additional data for the 2.1 and 2.2 ppm challenges in response to comments from Reviewer 1. Additional details are in Supporting Information.*

Linea 251: "dispersion modelling reported in Section 3.2 suggested that the ability to detect mixing ratio enhancements of ~1 ppm". Perhaps the Tildas measurements could be compared to the model results (here an example of polar plots that may help https://www.sciencedirect.com/science/article/pii/S1352231016307166)
*Response: Comparison of observed concentrations to on-site emissions, which were variable and subject to significant uncertainty, was beyond the scope of this manuscript.*

Linea 253: "Representative results for one month of data are shown in Figure 8". Please show the full comparison time series and show the regression plots.
*Response: This extensive information is available in the Supporting Material.*

Line 256-260: the slopes and R2 are mentioned, but the time series and regression plots are not shown, which are super useful to understand sensors' performance. Including other metrics like RMSE and MAE can also provide useful information.
*Response: This extensive information is available in the Supporting Material.*

Line 258: R2 is not a correlation coefficient, it is the Coefficient of Determination.
*Response: We define this term in the paper when it is first used.*

Line 262: "relative to baseline" is twice.
*Response: That was intended to convey the meaning.*

Figure 7: improved quality is needed (labels, labels, fonts, etc. include date and time). Extend it to the entire comparison period.
*Response: Our versions of the Figures appear to be of good resolution; if this is an issue in Figure conversion, we will address this with the publisher.*

Figure 8: improved quality is needed (labels, fonts, etc.) Merge the Tildas and Sensor x layouts into a single panel to compare the histograms easily. Use the entire comparison period.
*Response: Our versions of the Figures appear to be of good resolution; if this is an issue in Figure conversion, we will address this with the publisher.*

Figure 9: improved quality is needed (labels, fonts, etc.). It is not clear which panel corresponds to which sensor.
*Response: Our versions of the Figures appear to be of good resolution; if this is an issue in Figure conversion, we will address this with the publisher.*

Figure 10: improved quality is needed (labels, labels, fonts, etc. include date and time). Use the entire comparison period.
*Response: Our versions of the Figures appear to be of good resolution; if this is an issue in Figure conversion, we will address this with the publisher. Summaries of the performance for the entire comparison period is available in Table 4 of the revised manuscript Table 3 in the original submission). Shorter time periods are provided in the Figures to improve resolution.*

Table 3: Consider moving to supplementary. R2 is not a correlation coefficient.
*Response: We believe it should be retained.*

Table 4: Consider moving to supplementary.
*Response: We believe it should be retained.*